# ITERATIVE GRAPH SELF-DISTILLATION

## ABSTRACT

How to discriminatively vectorize graphs is a fundamental challenge that attracts increasing attentions in recent years. Inspired by the recent success of unsupervised contrastive learning, we aim to learn graph-level representation in an unsupervised manner. Specifically, we propose a novel unsupervised graph learning paradigm called Iterative Graph Self-Distillation (IGSD) which iteratively performs the teacher-student distillation with graph augmentations. Different from conventional knowledge distillation, IGSD constructs the teacher with an exponential moving average of the student model and distills the knowledge of itself. The intuition behind IGSD is to predict the teacher network representation of the graph pairs under different augmented views. As a natural extension, we also apply IGSD to semi-supervised scenarios by jointly regularizing the network with both supervised and unsupervised contrastive loss. Finally, we show that finetuning the IGSD-trained models with self-training can further improve the graph representation power. Empirically, we achieve significant and consistent performance gain on various graph datasets in both unsupervised and semi-supervised settings, which well validates the superiority of IGSD.

## 1 INTRODUCTION

Graphs are ubiquitous representations encoding relational structures across various domains. Learning low-dimensional vector representations of graphs is critical in various domains ranging from social science (Newman & Girvan, 2004) to bioinformatics (Duvenaud et al., 2015; Zhou et al., 2020). Many graph neural networks (GNNs) (Gilmer et al., 2017; Kipf & Welling, 2016; Xu et al., 2018) have been proposed to learn node and graph representations by aggregating information from every node's neighbors via non-linear transformation and aggregation functions. However, the key limitation of existing GNN architectures is that they often require a huge amount of labeled data to be competitive but annotating graphs like drug-target interaction networks is challenging since it needs domain-specific expertise. Therefore, unsupervised learning on graphs has been long studied, such as graph kernels (Shervashidze et al., 2011) and matrix-factorization approaches (Belkin & Niyogi, 2002).

Inspired by the recent success of unsupervised representation learning in various domains like images (Chen et al., 2020b; He et al., 2020) and texts (Radford et al., 2018), most related works in the graph domain either follow the pipeline of unsupervised pretraining (followed by fine-tuning) or *InfoMax principle* (Hjelm et al., 2018). The former often needs meticulous designs of pretext tasks (Hu et al., 2019; You et al., 2020) while the latter is dominant in unsupervised graph representation learning, which trains encoders to maximize the mutual information (MI) between the representations of the global graph and local patches (such as subgraphs) (Veličković et al., 2018; Sun et al., 2019; Hassani & Khasahmadi, 2020). However, MI-based approaches usually need to sample subgraphs as local views to contrast with global graphs. And they usually require an additional discriminator for scoring local-global pairs and negative samples, which is computationally prohibitive (Tschannen et al., 2019). Besides, the performance is also very sensitive to the choice of encoders and MI estimators (Tschannen et al., 2019). Moreover, MI-based approaches cannot be handily extended to the semi-supervised setting since local subgraphs lack labels that can be utilized for training. Therefore, we are seeking an approach that learns the entire graph representation by contrasting the whole graph directly without the need of MI estimation, discriminator and subgraph sampling.

Motivated by recent progress on contrastive learning, we propose the Iterative Graph Self-Distillation (IGSD), a teacher-student framework to learn graph representations by contrasting graph instances directly. The high-level idea of IGSD is based on graph contrastive learning where we pull sim-

ilar graphs together and push dissimilar graph away. However, the performance of conventional contrastive learning largely depends on how negative samples are selected. To learn discriminative representations and avoid collapsing to trivial solutions, a large set of negative samples (He et al., 2020; Chen et al., 2020b) or a special mining strategy (Schroff et al., 2015; He et al., 2020) are necessary. In order to alleviate the dependency on negative samples mining and still be able to learn discriminative graph representations, we propose to use self-distillation as a strong regularization to guide the graph representation learning.

In the IGSD framework, graph instances are augmented as several views to be encoded and projected into a latent space where we define a similarity metric for consistency-based training. The parameters of the teacher network are iteratively updated as an exponential moving average of the student network parameters, allowing the knowledge transfer between them. As merely small amount of labeled data is often available in many real-world applications, we further extend IGSD to the semi-supervised setting such that it can effectively utilize graph-level labels while considering arbitrary amounts of positive pairs belonging to the same class. Moreover, in order to leverage the information from pseudo-labels with high confidence, we develop a self-training algorithm based on the supervised contrastive loss for fine-tuning.

We experiment with real-world datasets in various scales and compare the performance of IGSD with state-of-the-art graph representation learning methods. Experimental results show that IGSD achieves competitive performance in both unsupervised and semi-supervised settings with different encoders and data augmentation choices. With the help of self-training, our performance can exceed state-of-the-art baselines by a large margin.

To summarize, we make the following contributions in this paper:

- We propose a self-distillation framework called IGSD for unsupervised graph-level representation learning where the teacher-student distillation is performed for contrasting graph pairs under different augmented views.
- We further extend IGSD to the semi-supervised scenario, where the labeled data are utilized effectively with the supervised contrastive loss and self-training.
- We empirically show that IGSD surpasses state-of-the-art methods in semi-supervised graph classification and molecular property prediction tasks and achieves performance competitive with state-of-the-art approaches in unsupervised graph classification tasks.

## 2 RELATED WORK

**Contrastive Learning** Modern unsupervised learning in the form of contrastive learning can be categorized into two types: context-instance contrast and context-context contrast (Liu et al., 2020). The context-instance contrast, or so-called global-local contrast focuses on modeling the belonging relationship between the local feature of a sample and its global context representation. Most unsupervised learning models on graphs like DGI (Veličković et al., 2018), InfoGraph (Sun et al., 2019), CMC-Graph (Hassani & Khasahmadi, 2020) fall into this category, following the *InfoMax principle* to maximize the the mutual information (MI) between the input and its representation. However, estimating MI is notoriously hard in MI-based contrastive learning and in practice tractable lower bound on this quantity is maximized instead. And maximizing tighter bounds on MI can result in worse representations without stronger inductive biases in sampling strategies, encoder architecture and parametrization of MI estimators (Tschannen et al., 2019). Besides, the intricacies of negative sampling in MI-based approaches impose key research challenges like improper amount of negative samples or biased negative sampling (Tschannen et al., 2019; Chuang et al., 2020). Another line of contrastive learning approaches called context-context contrast directly study the relationships between the global representations of different samples as what metric learning does. For instance, a recently proposed model BYOL (Grill et al., 2020) bootstraps the representations of the whole images directly. Focusing on global representations between samples and corresponding augmented views also allows instance-level supervision to be incorporated naturally like introducing supervised contrastive loss (Khosla et al., 2020) into the framework for learning powerful representations. Graph Contrastive Coding (GCC) (Qiu et al., 2020) is a pioneer to leverage instance discrimination as the pretext task for structural information pre-training. However, our work is fundamentally different from theirs. GCC focuses on structural similarity to find common and transferable structural

patterns across different graph datasets and the contrastive scheme is done through subgraph instance discrimination. On the contrary, our model aims at learning graph-level representation by directly contrasting graph instances such that data augmentation strategies and graph labels can be utilized naturally and effectively.

**Knowledge Distillation** Knowledge distillation (Hinton et al., 2015) is a method for transferring knowledge from one architecture to another, allowing model compression and inductive biases transfer. Self-distillation (Furlanello et al., 2018) is a special case when two architectures are identical, which can iteratively modify regularization and reduce over-fitting if perform suitable rounds (Mobahi et al., 2020). However, they often focus on closing the gap between the predictive results of student and teacher rather than defining similarity loss in latent space for contrastive learning.

**Semi-supervised Learning** Modern semi-supervised learning can be categorized into two kinds: multi-task learning and consistency training between two separate networks. Most widely used semi-supervised learning methods take the form of multi-task learning: $\arg\min_\theta \mathcal{L}_l(D_l, \theta) + w\mathcal{L}_u(D_u, \theta)$ on labeled data $D_l$ and unlabeled data $D_u$. By regularizing the learning process with unlabeled data, the decision boundary becomes more plausible. Another mainstream of semi-supervised learning lies in introducing student network and teacher network and enforcing consistency between them (Tarvainen & Valpola, 2017; Miyato et al., 2019; Lee, 2013). It has been shown that semi-supervised learning performance can be greatly improved via unsupervised pre-training of a (big) model, supervised fine-tuning on a few labeled examples, and distillation with unlabeled examples for refining and transferring the task-specific knowledge (Chen et al., 2020c). However, whether task-agnostic self-distillation would benefit semi-supervised learning is still underexplored.

## 3 PRELIMINARIES

### 3.1 FORMULATION

**Unsupervised Graph Representation Learning** Given a set of unlabeled graphs $\mathcal{G} = \{G_i\}_{i=1}^N$, we aim at learning the low-dimensional representation of every graph $G_i \in \mathcal{G}$ favorable for downstream tasks like graph classification.

**Semi-supervised Graph Representation Learning** Consider a whole dataset $\mathcal{G} = \mathcal{G}_L \cup \mathcal{G}_U$ composed by labeled data $\mathcal{G}_L = \{(G_i, y_i)\}_{i=1}^l$ and unlabeled data $\mathcal{G}_U = \{G_i\}_{i=l+1}^{l+u}$ (usually $u \gg l$), our goal is to learn a model that can make predictions on graph labels for unseen graphs. And with $K$ augmentations, we get $\mathcal{G}'_L = \{(G'_k, y'_k)\}_{k=1}^{Kl}$ and $\mathcal{G}'_U = \{G'_k\}_{k=l+1}^{K(l+u)}$ as our training data.

### 3.2 GRAPH REPRESENTATION LEARNING

We represent a graph instance as $G(\mathcal{V}, \mathcal{E})$ with the node set $\mathcal{V}$ and the edge set $\mathcal{E}$. The dominant ways of graph representation learning are graph neural networks with neural message passing mechanisms (Hamilton et al., 2017): for every node $v \in \mathcal{V}$, node representation $\mathbf{h}_v^k$ is iteratively computed from the features of their neighbor nodes $\mathcal{N}(v)$ using a differentiable aggregation function. Specifically, at the iteration $k$ we get the node embedding as:

$$\mathbf{h}_v^k = \sigma\left(\mathbf{W}^k \cdot \text{CONCAT}\left(\mathbf{h}_v^{k-1}, \text{AGGREGATE}_k\left(\{\mathbf{h}_u^{k-1}, \forall u \in \mathcal{N}(v)\}\right)\right)\right) \tag{1}$$

Then the graph-level representations can be attained by aggregating all node representations using a readout function like summation or set2set pooling (Vinyals et al., 2015).

### 3.3 GRAPH DATA AUGMENTATION

It has been shown that the learning performance of GNNs can be improved via graph diffusion, which serves as a homophily-based denoising filter on both features and edges in real graphs (Klicpera et al., 2019). The transformed graphs can also serve as effective augmented views in contrastive learning (Hassani & Khasahmadi, 2020). Inspired by that, we transform a graph $G$ with transition matrix $\boldsymbol{T}$ via graph diffusion and sparsification $\mathbf{S} = \sum_{k=0}^{\infty} \theta_k \boldsymbol{T}^k$ into a new graph with adjacency matrix $\mathbf{S}$ as an augmented view in our framework. While there are many design choices in coefficients $\theta_k$ like heat kernel, we employ Personalized PageRank (PPR) with $\theta_k^{PPR} = \alpha(1-\alpha)^k$ due to its

superior empirical performance (Hassani & Khasahmadi, 2020). As another augmentation choice, we randomly remove edges of graphs to attain corrupted graphs as augmented views to validate the robustness of models to different augmentation choices.

## 4 ITERATIVE GRAPH SELF-DISTILLATION

Intuitively, the goal of contrastive learning on graphs is to learn graph representations that are close in the metric space for positive pairs (graphs with the same labels) and far between negative pairs (graphs with different labels). To achieve this goal, IGSD employs the teacher-student distillation to iteratively refine representations by contrasting latent representations embedded by two networks and using additional predictor and EMA update to avoid collapsing to trivial solutions. Overall, IGSD encourages the closeness of augmented views from the same graph instances while pushing apart the representations from different ones.

### 4.1 ITERATIVE GRAPH SELF-DISTILLATION FRAMEWORK

In IGSD, we introduce a teacher-student architecture comprises two networks in similar structure composed by encoder $f_\theta$, projector $g_\theta$ and predictor $h_\theta$. We denote the components of the teacher network and the student network as $f_{\theta'}$, $g_{\theta'}$ and $f_\theta$, $g_\theta$, $h_\theta$ respectively.

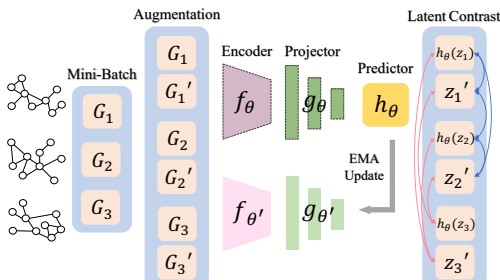

The overview of IGSD is illustrated in Figure 1. In IGSD, the procedure of contrastive learning on negative pairs is described as follows: we first augment the original input graphs $G_j$ to get augmented view(s) $G_j'$. Then $G_j'$ and different graph instance $G_i$ are fed respectively into two encoders $f_\theta$, $f_{\theta'}$ for extracting graph representations $\boldsymbol{h}, \boldsymbol{h}' = f_\theta(G_i), f_{\theta'}(G_j')$ with iterative message passing in Eqn. (1) and readout functions. The following projectors $g_\theta$, $g_{\theta'}$ transform graph representations to projections $\boldsymbol{z}, \boldsymbol{z}'$ via $\boldsymbol{z} = g_\theta(\boldsymbol{h}) = W^{(2)}\sigma(W^{(1)}\boldsymbol{h})$ and $\boldsymbol{z}' = g_{\theta'}(\boldsymbol{h}') = W'^{(2)}\sigma(W'^{(1)}\boldsymbol{h}')$, where $\sigma$ denotes a ReLU nonlinearity[1]. To prevent collapsing into a trivial solution (Grill et al., 2020), a specialized predictor is used in the student network for attaining the prediction $h_\theta(\boldsymbol{z}) = W_h^{(2)}\sigma(W_h^{(1)}\boldsymbol{z})$ of the projection $\boldsymbol{z}$. For positive pairs, we follow the same procedure except feeding the original and augmented view of the same graph into two networks respectively.

Figure 1: **Overview of IGSD**. Illustration of our framework in the case where we augment input graphs $G$ once to get $G'$ for only one forward pass. Blue and red arrows denote contrast on positive and negative pairs respectively.

To contrast latents $h_\theta(\boldsymbol{z})$ and $\boldsymbol{z}'$, we use $L_2$ norm in the latent space to approximate the semantic distance in the input space and the consistency loss can be defined as the mean square error between the normalized prediction $h_\theta(\boldsymbol{z})$ and projection $\boldsymbol{z}'$. By passing two graph instances $G_i$ and $G_j$ symmetrically, we can obtain the overall consistency loss:

$$\mathcal{L}^{\text{con}}(G_i, G_j) = \left\| h_\theta\left(\boldsymbol{z_i}\right) - \boldsymbol{z_j'} \right\|_2^2 + \left\| h_\theta\left(\boldsymbol{z_i'}\right) - \boldsymbol{z_j} \right\|_2^2 \tag{2}$$

With the consistency loss, the teacher network provides a regression target to train the student network, and its parameters $\theta'$ are updated as an exponential moving average (EMA) of the student parameters $\theta$ after weights of the student network have been updated using gradient descent:

$$\theta_t' \leftarrow \tau\theta_{t-1}' + (1-\tau)\theta_t \tag{3}$$

With the above iterative self-distillation procedure, we can aggregate information for averaging model weights over each training step instead of using the final weights directly (Athiwaratkun et al., 2018). It should be noted that maintaining a slow-moving average network is also employed in some models like MoCo (He et al., 2020) with different motivations: MoCo uses an EMA of encoder

---

[1]Although IGSD could directly predict the representations without projections, previous contrastive learning work (Chen et al., 2020b) in the image domain has shown that using projections improves performance empirically. We include the experimental results to validate the effects of projectors in Appendix A.3

and momentum encoder to update the encoder, ensuring the consistency of dictionary keys in the memory bank. On the other hand, IGSD uses a moving average network to produce prediction targets, enforcing the consistency of teacher and student for training the student network.

## 4.2 Unsupervised Learning with IGSD

In IGSD, to contrast the anchor $G_i$ with other graph instances $G_j$ (i.e. negative samples), we employ the following unsupervised InfoNCE objective (Oord et al., 2018):

$$\mathcal{L}^{\text{unsup}} = -\mathbb{E}_{G_i \sim \mathcal{G}} \left[ \log \frac{\exp\left(-\mathcal{L}^{\text{con}}(G_i, G_i)\right)}{\exp\left(-\mathcal{L}^{\text{con}}(G_i, G_i)\right) + \sum_{j=1}^{N-1} \mathbb{I}_{i \neq j} \cdot \exp\left(-\mathcal{L}^{\text{con}}(G_i, G_j)\right)} \right] \quad (4)$$

At the inference time, as semantic interpolations on samples, labels and latents result in better representations and can improve learning performance greatly (Zhang et al., 2017; Verma et al., 2019; Berthelot et al., 2019), we obtain the graph representation $\tilde{h}$ by interpolating the latent representations $h = f_\theta(G)$ and $h' = f_{\theta'}(G)$ with Mixup function $\text{Mix}_\lambda(a, b) = \lambda \cdot a + (1 - \lambda) \cdot b$:

$$\tilde{h} = \text{Mix}_\lambda(h, h') \quad (5)$$

## 4.3 Semi-supervised Learning with IGSD

To bridge the gap between unsupervised pretraining and downstream tasks, we extend our model to the semi-supervised setting. In this scenario, it is straightforward to plug in the unsupervised loss as a regularizer for representation learning. However, the instance-wise supervision limited to standard supervised learning may lead to biased negative sampling problems (Chuang et al., 2020). To tackle this challenge, we can use a small amount of labeled data further to generalize the similarity loss to handle arbitrary number of positive samples belonging to the same class:

$$\mathcal{L}^{\text{supcon}} = \sum_{i=1}^{Kl} \frac{1}{K N_{y_i'}} \sum_{j=1}^{Kl} \mathbb{I}_{i \neq j} \cdot \mathbb{I}_{y_i' = y_j'} \cdot \mathcal{L}^{\text{con}}(G_i, G_j) \quad (6)$$

where $N_{y_i'}$ denotes the total number of samples in the training set that have the same label $y_i'$ as anchor $i$. Thanks to the graph-level contrastive nature of IGSD, we are able to alleviate the biased negative sampling problems (Khosla et al., 2020) with supervised contrastive loss, which is crucial (Chuang et al., 2020) but unachievable in most MI-based contrastive learning models since subgraphs are generally hard to assign labels to. Besides, with this loss we are able to fine-tune our model effectively using self-training where pseudo-labels are assigned iteratively to unlabeled data.

With the standard supervised loss like cross entropy or mean square error $\mathcal{L}(\mathcal{G}_L, \theta)$, the overall objective can be summarized as:

$$\mathcal{L}^{\text{semi}} = \mathcal{L}(\mathcal{G}_L, \theta) + w\mathcal{L}^{\text{unsup}}(\mathcal{G}_L \cup \mathcal{G}_U, \theta) + w'\mathcal{L}^{\text{supcon}}(\mathcal{G}_L, \theta) \quad (7)$$

Common semi-supervised learning methods use consistency regularization to measure discrepancy between predictions made on perturbed unlabeled data points for better prediction stability and generalization (Oliver et al., 2018). By contrast, our methods enforce consistency constraints between latents from different views, which acts as a regularizer for learning directly from labels.

Labeled data provides extra supervision about graph classes and alleviates biased negative sampling. However, they are costly to attain in many areas. Therefore, we develop a contrastive self-training algorithm to leverage label information more effectively than cross entropy in the semi-supervised scenario. In the algorithm, we train the model using a small amount of labeled data and then fine-tune it by iterating between assigning pseudo-labels to unlabeled examples and training models using the augmented dataset. In this way, we harvest massive pseudo-labels for unlabeled examples.

With increasing size of the augmented labeled dataset, the discriminative power of IGSD can be improved iteratively by contrasting more positive pairs belonging to the same class. In this way, we accumulate high-quality pseudo-labels after each iteration to compute the supervised contrastive loss in Eqn. (6) and make distinction from conventional self-training algorithms (Rosenberg et al., 2005). On the other hand, traditional self-training can use psuedo-labels for computing cross entropy only.

## 5 EXPERIMENTS

### 5.1 EXPERIMENTAL SETUP

**Evaluation Tasks.** We conduct experiments by comparing with state-of-the-art models on three tasks. In **graph classification** tasks, we experiment in both the **unsupervised setting** where we only have access to all unlabeled samples in the dataset and the **semi-supervised setting** where we use a small fraction of labeled examples and treat the rest as unlabeled ones by ignoring their labels. In **molecular property prediction** tasks where labels are expensive to obtain, we only consider the **semi-supervised setting**.

**Datasets.** For graph classification tasks, we employ several widely-used graph kernel datasets (Kersting et al., 2016) for learning and evaluation: 3 bioinformatics datasets (MUTAG, PTC, NCI1) and 3 social network datasets (COLLAB, IMDB-B, IMDB-M) with statistics summarized in Table 1. In the semi-supervised graph regression tasks, we use the QM9 dataset containing 134,000 drug-like organic molecules (Ramakrishnan et al., 2014) with 9 heavy atoms and select the first ten physicochemical properties as regression targets for training and evaluation. For detailed description of the properties in the QM9 dataset, see the Appendix C of (Sun et al., 2019).

**Baselines.** In the unsupervised graph classification, we compare with the following representative baselines: CMC-Graph (Hassani & Khasahmadi, 2020), InfoGraph (Sun et al., 2019), GCC (Qiu et al., 2020), Graph2Vec (Narayanan et al., 2017) and Graph Kernels including Random Walk Kernel (Gärtner et al., 2003), Shortest Path Kernel (Kashima et al., 2003), Graphlet Kernel (Shervashidze et al., 2009), Weisfeiler-Lehman Sub-tree Kernel (WL SubTree) (Shervashidze et al., 2011), Deep Graph Kernels (Yanardag & Vishwanathan, 2015), Multi-Scale Laplacian Kernel (MLG) (Kondor & Pan, 2016) and Graph Convolutional Kernel Network (GCKN) (Chen et al., 2020a).

For the semi-supervised graph classification, we compare our method with competitive baselines like InfoGraph, InfoGraph* and Mean Teachers. And the GIN baseline doesn't have access to the unlabeled data. In the semi-supervised molecular property prediction tasks, baselines include InfoGraph, InfoGraph* and Mean Teachers (Tarvainen & Valpola, 2017).

**Model Configuration.** In our framework, We use GCNs (Kipf & Welling, 2016) and GINs (Xu et al., 2018) as encoders to attain node representations for unsupervised and semi-supervised graph classification respectively. For semi-supervised molecular property prediction, we employ message passing neural networks (MPNNs) (Gilmer et al., 2017) as backbone encoders to encode molecular graphs with rich edge attributes. All projectors and predictors are implemented as two-layer MLPs. For more details on hyper-parameters selection, refer to appendix A.2

In semi-supervised molecular property prediction tasks, we generate multiple views based on edge attributes (bond types) of rich-annotated molecular graphs for improving performance. Specifically, we perform label-preserving augmentation to attain multiple diffusion matrixes of every graph on different edge attributes while ignoring others respectively. The diffusion matrix gives a denser graph based on each type of edges to leverage edge features better. We train our models using different numbers of augmented training data and select the amount using cross validation.

For unsupervised graph classification, we adopt LIB-SVM (Chang & Lin, 2011) with $C$ parameter selected in $\{1e-3, 1e-2, \ldots, 1e2, 1e3\}$ as our downstream classifier. Then we use 10-fold cross validation accuracy as the classification performance and repeat the experiments 5 times to report the mean and standard deviation. For semi-supervised graph classification, we randomly select $5\%$ of training data as labeled data while treat the rest as unlabeled one and report the best test set accuracy in 300 epochs. Following the experimental setup in (Sun et al., 2019), we randomly choose 5000, 10000, 10000 samples for training, validation and testing respectively and the rest are treated as unlabeled training data for the molecular property prediction tasks.

### 5.2 NUMERICAL RESULTS

**Results on unsupervised graph classification.** We first present the results of the unsupervised setting in Table 1. All graph kernels give inferior performance except in the PTC dataset. The Random Walk kernel runs out of memory and the Multi-Scale Laplacian Kernel suffers from a long running time (exceeds 24 hours) in two larger datasets. IGSD outperforms state-of-the-art baselines

| | Datasets | MUTAG | IMDB-B | IMDB-M | NCI1 | COLLAB | PTC |
|---|---|---|---|---|---|---|---|
| **Datasets** | # graphs | 188 | 1000 | 1500 | 4110 | 5000 | 344 |
| | # classes | 2 | 2 | 3 | 2 | 3 | 2 |
| | Avg # nodes | 17.9 | 19.8 | 13.0 | 29.8 | 74.5 | 25.5 |
| **Graph Kernels** | Random Walk | $83.7 \pm 1.5$ | $50.7 \pm 0.3$ | $34.7 \pm 0.2$ | OMR | OMR | $57.9 \pm 1.3$ |
| | SHORTEST PATH | $85.2 \pm 2.4$ | $55.6 \pm 0.2$ | $38.0 \pm 0.3$ | $51.3 \pm 0.6$ | $49.8 \pm 1.2$ | $58.2 \pm 2.4$ |
| | GRAPHLET KERNEL | $81.7 \pm 2.1$ | $65.9 \pm 1.0$ | $43.9 \pm 0.4$ | $53.9 \pm 0.4$ | $56.3 \pm 0.6$ | $57.3 \pm 1.4$ |
| | WL subtree | $80.7 \pm 3.0$ | $72.3 \pm 3.4$ | $47.0 \pm 0.5$ | $55.1 \pm 1.6$ | $50.2 \pm 0.9$ | $58.0 \pm 0.5$ |
| | DEEP GRAPH | $87.4 \pm 2.7$ | $67.0 \pm 0.6$ | $44.6 \pm 0.5$ | $54.5 \pm 1.2$ | $52.1 \pm 1.0$ | $60.1 \pm 2.6$ |
| | MLG | $87.9 \pm 1.6$ | $66.6 \pm 0.3$ | $41.2 \pm 0.0$ | $>1$ Day | $>1$ Day | $\mathbf{63.3 \pm 1.5}$ |
| | GCKN | $87.2 \pm 6.8$ | $70.5 \pm 3.1$ | $50.8 \pm 0.8$ | $70.6 \pm 2.0$ | $54.3 \pm 1.0$ | $58.4 \pm 7.6$ |
| **Unsupervised** | GRAPH2VEC | $83.2 \pm 9.6$ | $71.1 \pm 0.5$ | $50.4 \pm 0.9$ | $73.2 \pm 1.8$ | $47.9 \pm 0.3$ | $60.2 \pm 6.9$ |
| | INFOGRAPH | $89.0 \pm 1.1$ | $74.2 \pm 0.7$ | $49.7 \pm 0.5$ | $73.8 \pm 0.7$ | $67.6 \pm 1.2$ | $61.7 \pm 1.7$ |
| | CMC-GRAPH | $89.7 \pm 1.1$ | $74.2 \pm 0.7$ | $51.2 \pm 0.5$ | $75.0 \pm 0.7$ | $68.9 \pm 1.9$ | $62.5 \pm 1.7$ |
| | GCC | $86.4 \pm 0.5$ | $71.9 \pm 0.5$ | $48.9 \pm 0.8$ | $66.9 \pm 0.2$ | $\mathbf{75.2 \pm 0.3}$ | $58.4 \pm 1.2$ |
| | OURS (RANDOM) | $85.7 \pm 2.1$ | $71.6 \pm 1.2$ | $49.2 \pm 0.6$ | $75.1 \pm 0.4$ | $65.8 \pm 1.0$ | $57.6 \pm 1.5$ |
| | OURS | $\mathbf{90.2 \pm 0.7}$ | $\mathbf{74.7 \pm 0.6}$ | $\mathbf{51.5 \pm 0.3}$ | $\mathbf{75.4 \pm 0.3}$ | $70.4 \pm 1.1$ | $61.4 \pm 1.7$ |

Table 1: **Graph classification accuracies (%) for kernels and unsupervised methods on 6 datasets.** We report the mean and standard deviation of final results with five runs. '>1 day' represents that the computation exceeds 24 hours. 'OMR' means out of memory error.

| Datasets | IMDB-B | IMDB-M | COLLAB | NCI1 |
|---|---|---|---|---|
| MEAN TEACHERS | 69.0 | 49.3 | 72.5 | 71.1 |
| INFOGRAPH* | 71.0 | 49.3 | 67.6 | 71.1 |
| GIN (SUPERVISED ONLY) | 67.0 | 50.0 | 71.4 | 67.9 |
| OURS (UNSUP) | 72.0 | 50.0 | 72.6 | 70.6 |
| OURS (SUPCON) | 75.0 | 52.0 | 73.4 | 67.9 |
| GIN (SUPERVISED ONLY)+SELF-TRAINING | 72.0 | 51.3 | 70.4 | 74.0 |
| OURS (UNSUP)+SELF-TRAINING | 73.0 | 54,0 | 71 | 72.5 |
| OURS (SUPCON)+SELF-TRAINING | **77.0** | **55.3** | **73.6** | **77.1** |

Table 2: **Graph classification accuracies (%) of semi-supervised experiments on 4 datasets.** We report the best results on test set in 300 epochs.

like InfoGraph, CMC-Graph and GCC in most datasets, showing that IGSD can learn expressive graph-level representations for downstream classifiers. Besides, our model still achieve competitive results in datasets like IMDB-M and NCI1 with random dropping augmentation, which demonstrates the robustness of IGSD with different choices of data augmentation strategies.

**Results on semi-supervised graph classification.** We further apply our model to semi-supervised graph classification tasks with results demonstrated in Table 4, where we set $w$ and $w'$ in Eqn. (7) to be 1 and 0 as Ours (Unsup) while 0 and 1 as Ours (SupCon). In this setting, our model performs better than Mean Teachers and InfoGraph*. Both the unsupervised loss and supervised contrastive loss provide extra performance gain compared with GIN using supervised data only. Besides, both of their performance can be improved significantly combined using self-training especially with supervised contrastive loss. It makes empirical sense since self-training iteratively assigns psuedo-labels with high confidence to unlabeled data, which provides extra supervision on their categories under contrastive learning framework.

**Results on semi-supervised molecular property prediction.** We present the regression performance of our model measured in the QM9 dataset in Figure 2. We display the performance of our model and baselines as mean square error ratio with respect to supervised results and our model outperforms all baselines in 9 out of 10 tasks compared with strong baselines InfoGraph, InfoGraph* and Mean Teachers. And in some tasks like R2 (5), U0 (7) and U (8), IGSD achieves significant performance gains against its counterparts, which demonstrates the ability to transfer knowledge learned from unsupervised data for supervised tasks.

## 5.3 ABLATION STUDIES AND ANALYSIS

**Effects of self-training.** We first investigate the effects of self-training for our model performance in table 4. Results show that self-training can improve the GIN baseline and our models with unsupervised loss (Unsup) or supervised contrastive loss (SupCon). The improvement is even more significant combined with supervised contrastive loss since high-quality pseudo-labels provide

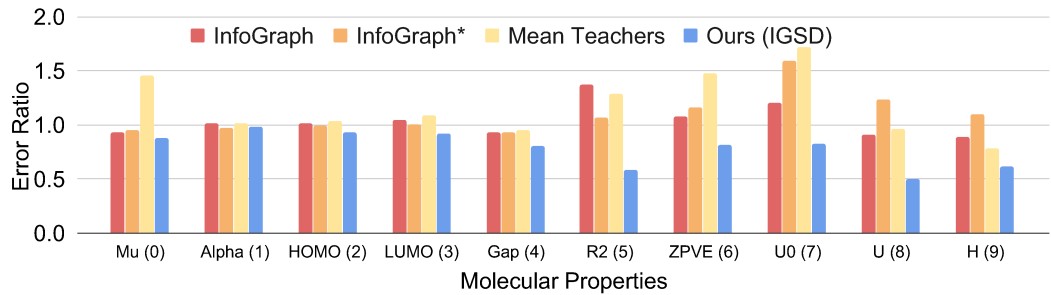

Figure 2: **Semi-supervised molecular property prediction results in terms of mean absolute error (MAE) ratio.** The histogram shows error ratio with respect to supervised results (1.0) of every semi-supervised models. Lower scores are better and a model outperforms the supervised baseline when the score is less than 1.0.

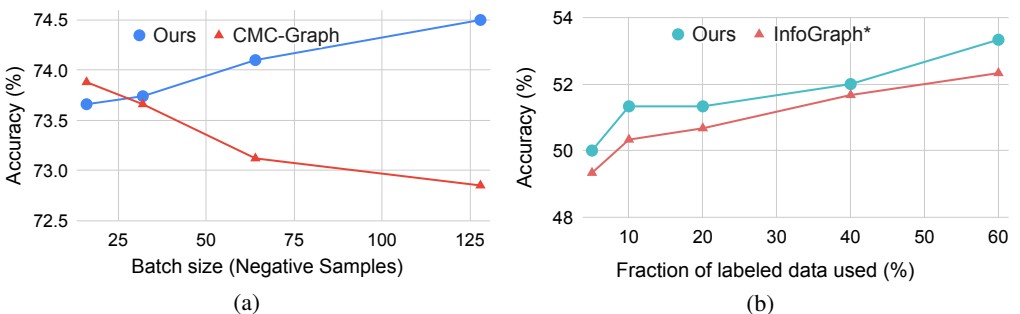

Figure 3: **Ablation studies.** (a) Unsupervised performance with different batch size (number of negative pairs); (b) Semi-supervised graph classification accuracy with different proportion of labeled data.

additional information of graph categories. Moreover, our self-training algorithm consistently outperforms the traditional self-training baseline, which further validates the superiority of our model.

**Effects of different amount of negative pairs.** We then conduct ablation experiments on the amount of negative pairs by varying batch size over $\{16, 32, 64, 128\}$ with results on IMDB-B dataset shown in Figure 3a. Both methods contrast negative pairs batch-wise and increasing batch size improves the performance of IGSD while degrades CMC-Graph. When batch size is greater than 32, IGSD outperforms CMC-Graph and the performance gap becomes larger as the batch size increases, which means IGSD is better at leveraging negative pairs for learning effective representations than CMC-Graph.

**Effects of different proportion of labeled data.** We also investigate the performance of different models with different proportion of labeled data with IMDB-B dataset. As illustrated in Figure 3b, IGSD outperforms strong InfoGraph* baseline given different amount of labeled data consistently. And the performance gain is most significant when the fraction of labeled data is $10\%$ since our models can leverage labels more effectively by regularizing original unsupervised learning objective when labels are scarce.

## 6 CONCLUSIONS

In this paper, we propose IGSD, a novel unsupervised graph-level representation learning framework via self-distillation. Our framework iteratively performs teach-student distillation by contrasting augmented views of graph instances. Experimental results in both unsupervised and semi-supervised settings show that IGSD is not only able to learn effective graph representations competitive with state-of-the-art models but also robust with choices of encoders and augmentation strategies. In the future, we plan to apply our framework to other graph learning tasks and investigate the design of view generators to generative effective views automatically.

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

# A   APPENDIX

## A.1   RELATED WORK

**Graph Representation Learning**   Traditionally, graph kernels are widely used for learning node and graph representations. This common process includes meticulous designs like decomposing graphs into substructures and using kernel functions like Weisfeiler-Leman graph kernel (Shervashidze et al., 2011) to measure graph similarity between them. However, they usually require non-trivial hand-crafted substructures and domain-specific kernel functions to measure the similarity while yields inferior performance on downstream tasks like node classification and graph classification. Moreover, they often suffer from poor scalability (Borgwardt & Kriegel, 2005) and great memory consumption (Kondor & Pan, 2016) due to some procedures like path extraction and recursive subgraph construction. Recently, there has been increasing interest in Graph Neural Network (GNN) approaches for graph representation learning and many GNN variants have been proposed (Ramakrishnan et al., 2014; Kipf & Welling, 2016; Xu et al., 2018). However, they mainly focus on supervised settings.

**Data augmentation**   Data augmentation strategies on graphs are limited since defining views of graphs is a non-trivial task. There are two common choices of augmentations on graphs (1) feature-space augmentation and (2) structure-space augmentation. A straightforward way is to corrupt the adjacency matrix which preserves the features but adds or removes edges from the adjacency matrix with some probability distribution (Veličković et al., 2018). Zhao et al. (2020) improves performance in GNN-based semi-supervised node classification via edge prediction. Empirical results show that diffusion matrix can serve as a denoising filter to augment graph data for improving graph representation learning significantly both in supervised (Klicpera et al., 2019) and unsupervised settings (Hassani & Khasahmadi, 2020). Hassani & Khasahmadi (2020) shows the benefits of treating diffusion matrix as an augmented view of mutual information-based contrastive graph representation learning. Attaining effective views is non-trivial since we need to consider factors like mutual information to preserve label information w.r.t the downstream task (Tian et al., 2020).

## A.2   HYPER-PARAMETERS

For hyper-parameter tuning, we select number of GCN layers over $\{2, 8, 12\}$, batch size over $\{16, 32, 64, 128, 256, 512\}$, number of epochs over $\{20, 40, 100\}$ and learning rate over $\{1e\text{-}4, 1e\text{-}3\}$ in unsupervised graph classification.

The hyper-parameters we tune for semi-supervised graph classification and molecular property prediction are the same in (Xu et al., 2018) and (Sun et al., 2019), respectively.

In all experiments, we fix the fixed $\alpha = 0.2$ for PPR graph diffusion and set the weighting coefficient of Mixup function to be 0.5 and tune our projection hidden size over $\{1024, 2048\}$ and projection size over $\{256, 512\}$. We start self-training after 30 epochs and tune the number of iterations over $\{20, 50\}$, pseudo-labeling threshold over $\{0.9, 0.95\}$.

## A.3   EFFECT OF PROJECTORS

While we could directly predict the representation $y$ and not a projection $z$, previous contrastive learning works in the image domain like (Chen et al., 2020b) have empirically shown that using this projection improves performance. We also further investigate the performance with and without the projector on 4 datasets:

| Datasets | MUTAG | IMDB-B | IMDB-M | PTC |
|---|---|---|---|---|
| IGSD | **90.2 ± 0.7** | **74.7 ± 0.6** | **51.5 ± 0.3** | **61.4 ± 1.7** |
| IGSD W/O PROJECTOR | 87.7 ± 0.9 | 74.2 ± 0.6 | 51.1 ± 0.6 | 56.7 ± 0.9 |

Table 3: **Effect of projectors on Graph classification accuracies (%).**

Results above show that dropping the projector degrades the performance, which indicates the necessity of a projector.

Meanwhile, to investigate the effect of projectors on model performance, we fix the output size of layers in encoders so that their output size is always 512. Then we conducted ablation experiments on different size of the projection head on IMDB-B with the following results:

| Size | 218 | 512 | 1024 | 2048 |
|------|-----|-----|------|------|
| IGSD | $74.5 \pm 0.6$ | $74.8 \pm 0.4$ | $74.7 \pm 0.6$ | $74.9 \pm 0.8$ |

Table 4: **Effect of projector hidden size on Graph classification accuracies (%).**

In general, the performance is insensitive to the projection size while a larger projection size could slightly improve the unsupervised learning performance.

