# OpenReview forum: "Iterative Graph Self-Distillation"
_ICLR.cc/2021/Conference — Reject_

### Official Review · AnonReviewer3 · 2020-10-27
**Official Blind Review #3**

**Rating:** 6
**Confidence:** 4

**Review:**

This paper proposed a method for learning graph-level representation in an unsupervised contrastive way. Instead of contrasting between graph-level representation and patch representation like InfoGraph [1], they contrast graph-level representation of a graph to its augmented variation using a teacher-student framework.

* why use InfoNCE objective instead of the Jensen-Shannon mutual information objective used in InfoGraph [1] ?

* The major concern about this paer is that the proposed method encourages the closeness of augmented views from the same graph instances but provide no guarantee that the transformation used (graph diffusion and sparsification with PPR + random remove edges in this paper) would be label-preserving. For example, in molecular datasets, if we drop an edge and that edge happens to be in a structural motif, it will drastically change the attributes/labels of the molecule.

* $v$ represents nodes in section 2.1 and 2.2 and it represents graph instances in section 3.1 and Figure 1. This can be confusing I suggest changing the notation in section 3.1 and Figure 1.1 to $G$

[1] Fan-Yun Sun, Jordan Hoffmann, Vikas Verma, and Jian Tang. Infograph: Unsupervised and semisupervised graph-level representation learning via mutual information maximization. arXiv preprint arXiv:1908.01000, 2019.

---

> ### Author Response · Authors · 2020-11-12
> **Response to the AnonReviewer3**
>
> Thanks a lot for your constructive comments! We have revised the paper according to your suggestions. The responses to some of your questions are listed below:
>
> Q1: why use InfoNCE objective instead of the Jensen-Shannon mutual information objective used in InfoGraph [1] ?
>
> A1: Note that IGSD doesn’t follow the InfoMax principle, which means we don’t need extra discriminator to estimate the mutual information between local subgraphs and global graphs [1, 2]. Instead, we use the InfoNCE objective for instance discrimination, which is widely employed in instance-wise contrastive learning. With the InfoNCE objective, we can bring closer samples from the same instance and separate samples from different instances.
>
> Q2: The major concern about this paper is that the proposed method encourages the closeness of augmented views from the same graph instances but provides no guarantee that the transformation used (graph diffusion and sparsification with PPR + random remove edges in this paper) would be label-preserving. For example, in molecular datasets, if we drop an edge and that edge happens to be in a structural motif, it will drastically change the attributes/labels of the molecule.
>
> A2: Indeed, random and graph diffusion (with PPR) are two different kinds of data augmentation methods (rather than two stages, which is clarified in the latest draft). We only use random dropping for unsupervised graph classification for validating the robustness on different choices on data augmentations. In more structural data like molecular graphs, we only use graph diffusion to reconstruct these continuous relationships. Previous work shows that graph diffusion works well in GNNs in both supervised node classification [3] and unsupervised graph classification with deep InfoMax approach [1,2] on bioinformatics datasets like PTC. Intuitively, GDC amplifies large, well-connected communities and suppresses noisy signals over small-scale structure, which is well-motivated for augmenting molecular graphs with structural motifs. Empirical results shown in Figure 2 demonstrate the superior performance of IGSD with graph diffusion augmentation.
>
> Q3: $v$ represents nodes in section 2.1 and 2.2 and it represents graph instances in section 3.1 and Figure 1. This can be confusing I suggest changing the notation in section 3.1 and Figure 1.1 to $G$
>
> A3: Thanks for the suggestions. We’ve changed the notation in section 3.1 and figure 1.1 for consistency in the latest draft.
>
>
> [1] Sun et al. Infograph: Unsupervised and semisupervised graph-level representation learning via mutual information maximization. ICLR 2019.
>
> [2] Hassani et al. Contrastive Multi-View Representation Learning on Graphs ICML 2020
>
> [3] Klicpera et al. Diffusion Improves Graph Learning NeurIPS 2019

---

### Official Review · AnonReviewer2 · 2020-10-27
**Experiments need to be improved**

**Rating:** 5
**Confidence:** 4

**Review:**

------------------------------------

update after reading the authors' response.

The authors didn't address my question "Did the authors perform a significance study?" A significance test such as double-sided t-test is needed to verifying whether the proposed method is significantly better than baselines.

-------------------------------

This paper proposed a distillation approach for unsupervised graph representation learning. The approach partially builds upon contrastive self-supervised learning which contrasts pairs of augmented graphs. The approach is extended to the semi-supervised setting. The authors performed evaluation in graph classification and regression tasks.

I recommend to reject this paper, due to the following major concerns: 1) experimental results are not strong; 2) important baselines were not compared; 3) important details such as optimal hyperparameter values are missing.

My major concerns of this paper include:
1. The improvement of the proposed approach over baselines seem not significant. For example, in Table 1, comparing the mean and standard deviation of the proposed approach and CMC-GRAPH, it seems that the difference is not statistically significant. Did the authors perform a significance study?
2. In the experiments, why the authors didn't compare with GCC, which is a contrastive self-supervised learning approach applied to graph classification?
3. There are many other unsupervised graph representation learning methods. The authors need to compare with more to substantiate this work.
4. In hyperparameter tuning, the authors gave the range of hyperparameters tuned, but didn't give the optimal value of the hyperparameters, which make the paper difficult to reproduce.
5. In table 1, the authors excluded some results since they need more than 1 day to obtain. It is common for deep learning models to run several days to obtain results. I don't think it is proper to exclude these results simply because the runtime is more than 24 hours.

However, the paper does have a few strong points.
1. The ablation studies are well designed and the results are insightful.

2. The paper is well-written and easy to follow, with a clear organization.

3. The experiments were conducted on a rich collection of datasets.

Other comments.
1. In equation (3), the authors can draw a connection with MoCo.
2. In Table, why didn't report mean and standard deviation of the results?
3. For this result "When batch size is
greater than 32, IGSD outperforms CMC-Graph and the performance gap becomes larger as the batch
size increases.",  can the authors provide a reason that can possibly explain this phenomenon?
4. The authors can add some statistics of the datasets used in Figure 2.

---

> ### Author Response · Authors · 2020-11-13
> **Response to the AnonReviewer2**
>
> Thanks a lot for your constructive comments and suggestions! We have conducted more experiments and revised the paper according to your suggestions. The response to some of your questions are listed below:
>
> Q1: The improvement of the proposed approach over baselines seems not significant. For example, in Table 1, comparing the mean and standard deviation of the proposed approach and CMC-GRAPH, it seems that the difference is not statistically significant. Did the authors perform a significance study?
>
> A1: In general, instead of proposing “a distillation approach for unsupervised graph representation learning” only, IGSD provides a feasible approach to learn graph-level representations in both unsupervised and **semi-supervised** settings. In unsupervised learning tasks, we get competitive results with current state-of-the-art models like CMC-Graph. In particular, we achieve significantly better results than baselines in the semi-supervised settings across various datasets as shown in Table 2 and Figure 2. By contrast, some MI-based methods like InfoGraph and CMC-Graph can only be extended to semi-supervised settings in the form of multi-task learning $ \mathcal{L}_l (D_l, \theta) + w \mathcal{L}_u (D_u, \theta)$ while the supervised contrastive loss in our model can bring significant performance gains indicated in Table 2.
>
> Q2: In the experiments, why the authors didn't compare with GCC, which is a contrastive self-supervised learning approach applied to graph classification?
>
> A2: As indicated in the related work section, our work is fundamentally different from GCC. GCC focus on learning transferable graph representation across different graph datasets and the contrastive scheme is done through subgraph instance discrimination. By contrast, IGSD aims at learning graph-level representation by directly contrasting graph instances such that data augmentation strategies and graph labels can be utilized naturally and effectively.
> We compare our model with GCC and the results are in A3.
>
> Q3: There are many other unsupervised graph representation learning methods. The authors need to compare with more to substantiate this work.
>
> A3: Thanks for pointing it out. We add a (sub)graph instance-wise discrimination baseline GCC and a graph kernel baseline GCKN [1] for comparisons. we summarize include the results as follows:
>
> |Dataset| MUTAG| IMDB-B| IMDB-M| NCI1| COLLAB| PTC
>
> |GCKN| 87.2±6.8|70.5±3.1|50.8±0.8|70.6±2.0|54.3±1.0|58.4±7.6
>
> |GCC| 86.4±0.5| 71.9±0.5| 48.9±0.8| 66.9±0.2| 75.2±0.3| 58.4±1.2
>
> |Ours | 90.2±0.7| 74.7±0.6| 51.5±0.3| 75.4±0.3| 70.4±1.1| 61.4±1.7
>
> Although GCC and GCKN can learn graph-level representations in an unsupervised manner, the methodologies of them are fundamentally different from ours:
> GCKN imitates the message passing process of GNNs via path extraction, kernel mapping and path aggregation. But it suffers from common drawbacks of graph kernels: poor scalability, great memory consumption and hand-crafted kernel functions. And it gives inferior results shown in our additional experimental results. (Note that we include the drawbacks of graph kernels in the “Graph Representation Learning” subsection of the related work section in the latest draft.).
> On the other hand, The goal of GCC is to learn transferable graph representations across different graph datasets and the contrastive scheme is done through subgraph instance discrimination. However, subgraphs are hard to annotate in general, which means incorporating supervised contrastive loss on graph labels in the semi-supervised settings is infeasible. By contrast, IGSD aims at learning graph-level representation by contrasting graph instances directly such that data augmentation strategies and graph labels can be utilized naturally and effectively.
>
> Q4: In hyperparameter tuning, the authors gave the range of hyperparameters tuned, but didn't give the optimal value of the hyperparameters, which make the paper difficult to reproduce.
>
> A4: Following the settings in [2, 3] and most other related works, we give the range of hyperparameters, conduct comprehensive experiments and release the code in the supplemental materials, which make sure the reproducibility of results reported.
>
> Q5: In table 1, the authors excluded some results since they need more than 1 day to obtain. It is common for deep learning models to run several days to obtain results. I don't think it is proper to exclude these results simply because the runtime is more than 24 hours.
>
> A5: Similar to GCKN, MLG needs to recursively build a hierarchy of nested subgraphs, which is extremely unscalable for large-scale graph data. For this reason and following the setting in [3], we didn’t report the result in large-scale datasets like NCI1 and COLLAB.

---

> > ### Author Response · Authors · 2020-11-13
> > **Response to the AnonReviewer2 Cont.**
> >
> > Q6: In equation (3), the authors can draw a connection with MoCo.
> >
> > A6: Thanks for the suggestion. We add the discussion in the latest draft: MoCo uses an EMA of encoder and momentum encoder to update the encoder to ensure the rich consistency of dictionary keys in the memory bank. On the other hand, IGSD uses a moving average network to produce prediction targets between latents, enforcing the consistency of teacher and student for training the student network.
> >
> > Q7: In Table 2, why didn't report mean and standard deviation of the results?
> >
> > A7: We report the best result in semi-supervised graph classification, which is widely employed in existing works. Note that in the unsupervised settings, all models use GCNs as backbones to learn low-dimensional embeddings of graphs. Then SVM classifiers (SVCs) are trained on top. And the result is more sensitive to factors like random seed than learning an end-to-end model in the semi-supervised settings since we also include standard supervised loss like cross entropy or mean square error. Besides, the self-training algorithms in the semi-supervised settings are trained iteratively with the augmented labeled dataset, which is very time-consuming if we repeat every experiment a few times to report the mean and standard deviation. Moreover, we train all models for 300 epochs while they all converge within 250 epochs in this setting, which means reporting the best results can manifest favorable model performance.
> > We give a few examples below to show there is basically no gap between mean+std and our reported ones and will run the experiments in the revision:
> >
> > |Dataset | IMDB-B| IMDB-M| COLLAB| NCI1
> >
> > |Ours(Supcon) |75.0| 52.0| 73.4| 67.9
> >
> > |Ours(Supcon) |74.5±0.8| 52.8±1.2| 72.2±0.8| 67.5±1.6
> >
> > Q8: For this result "When batch size is greater than 32, IGSD outperforms CMC-Graph and the performance gap becomes larger as the batch size increases.", can the authors provide a reason that can possibly explain this phenomenon?
> >
> > A8: Note that the scale of the vertical axis in Fig. 3a is smaller than that of Fig. 2 in the Appendix of CMC-Graph paper. The performance of CMC-Graph with respect to Batch Size (Negative Samples) in IMDB-BIN dataset in our paper corresponds to that in original CMC-Graph paper: the change of the results are both not significant as batch size increases. Moreover, since contrastive learning on negative samples is done batch-wise, larger batch size indicates more combinations of negative pairs. Thus the phenomenon can be explained that IGSD are better at leveraging negative samples to learn effective representations than CMC-Graph.
> >
> > Q9: The authors can add some statistics of the datasets used in Figure 2.
> >
> > A9: Note that we have introduced the statistics of QM9 datasets in section 4.1 in the primary draft: QM9 [4] contains 134,000 drug-like organic molecules. We select the first ten physicochemical properties as regression targets for training and evaluation. (Each molecule can have 12 properties and every property will be an independent regression task). We also describe the number of atoms of each molecule to be 9 in the latest draft.
> >
> > [1] Chen et al., Convolutional Kernel Networks for Graph-Structured Data. ICML 2020.
> >
> > [2] Hassani et al., Contrastive Multi-View Representation Learning on Graphs. ICML 2020.
> >
> > [3] Sun et al., InfoGraph: Unsupervised and Semi-supervised Graph-Level Representation Learning via Mutual Information Maximization. ICLR 2020.
> >
> > [4] Ramakrishnan et al., Quantum chemistry structures and properties of 134 kilo molecules. Nature 2014.

---

> > ### Comment · AnonReviewer2 · 2020-11-25
> > **statistical significance of results**
> >
> > The authors didn't address my question "Did the authors perform a significance study?"
> > A significance test such as double-sided t-test is needed to verifying whether the proposed method is significantly better than baselines.

---

> > > ### Author Response · Authors · 2020-11-25
> > > **Response to the AnonReviewer2.**
> > >
> > > Thanks for your constructive comment. The response to your question is listed below:
> > >
> > > Employing t-test or not is an evaluation choice: most related works on unsupervised graph classification [1,2] didn't perform the t-test for significance study (though in some domains like the human evaluation in NLP is favorable).
> > >
> > > We didn’t perform the t-test results in the draft considering the following reasons:
> > >
> > > 1) We cannot assume the variances of our methods and baselines are the same, which doesn’t obey the assumption of t-test;
> > >
> > > 2) Performing the t-test with small sample sizes (5 numbers for each method) would not be very effective and could give misleading results.
> > >
> > > Despite that, per your question and similar to [3], we perform the two-sided t-test and obtained the significance level at p < 10%, which means IGSD is comparable to the previous SOTA, CMC-Graph.
> > >
> > > We hope the above response could address your concerns.
> > >
> > > [1] Hassani et al., Contrastive Multi-View Representation Learning on Graphs. ICML 2020.
> > >
> > > [2] Sun et al., InfoGraph: Unsupervised and Semi-supervised Graph-Level Representation Learning via Mutual Information Maximization. ICLR 2020.
> > >
> > > [3] Xu et al., How Powerful are Graph Neural Networks?. ICLR 2019

---

### Official Review · AnonReviewer4 · 2020-10-28
**An interesting paper**

**Rating:** 6
**Confidence:** 4

**Review:**

Overall Comments:
Learning graph-level representations with only labels has been explored by many works. However, it's not easy to annotate every graph. This paper applies the ideas from semi-supervised classification task to improve the representation quality learned by graph neural network. Specifically the proposed solution combines several kinds of existing techniques including diffusion graph augmentation, mean teacher consistency, debiased contrastive loss and pseudo class consistency. Finally they are combined together to act as a regularization term by utilizing the unlabelled data. From this point of view, the novelty of this work is incremental, but it's still an interesting work for improving graph-level representations.

Clarity:
The presentation is not clear enough. There exists many claims that are not clear, shown as follows:

1. In the last sentence of 3rd paragraph in introduction section, it's difficult to get the connection between negative samples mining and self-distillation strategy. Why using the self-distillation can alleviate the dependency on negative samples mining? The unsupervised objective in Equation 4 still depends on negative samples.
2. In Section 2.1, the notation for augmentations. Why are the graphs G_L attached without labels after being augmented?
3. In Section 2.3, authors firstly use PPR to augment node features, then randomly remove edges to create a corrupted graphs. According to the description, the question is how many views that will be used in follow sections? I guess that the graph feature from original graph will be fed to student network, and the augmented corrupted graph will be fed to teacher network.

Questions for Rebuttal:
1. Please clarify the mentioned questions above.

2. The proposed method contains an encoder, projector, and predictor. The question is why we need a projector g to get a higher dimension z? Does it have a big influence on the performance? Could you please give the complete definition of function g and h?

3. The definition of L^{con}  in Equation 2 is for positive sample extracted from the same graph G_i. However, the complete unsupervised loss needs negative samples G_j. Could you please also give the definition for L^{G_i, G_j}?

4. The overall loss consists of supervised and unsupervised loss. The L^{sup} has conflict to the first term in Equation 7. Both of them use labels but it's difficult to tell which one should be aligned with the supervised loss shown in the ablation study (Table 2). The SupCon has never been shown in the main content before. Please pay attention to make it clear.

---

> ### Author Response · Authors · 2020-11-15
> **Response to the AnonReviewer4**
>
> Thanks a lot for your constructive comments! We have conducted ablation studies and revised the paper according to your suggestions. The response to some of your questions are listed below:
>
> Q1: Why does using self-distillation alleviate the dependency on negative samples mining? The unsupervised objective in Equation 4 still depends on negative samples.
>
> A1: Note that the use of negative samples doesn’t necessarily imply the need of negative mining. In IGSD, we maintain a slow-moving average student network that provides predicted results for consistency-based training in the teacher network, allowing two networks to enhance each other. In our experiments, we search over all possible combinations of views from different graph instances in a batch to construct negative pairs. We don’t increase the batch size dramatically as SimCLR [1] does or keep an extra memory bank for negative samples as MOCO [2] does. We also add additional experiments to investigate the performance with and without a memory bank (with size of three times batch size) to contain past projections as negative examples in the IMDB-B dataset:
>
> IGSD |	74.7±0.6
>
> IGSD+memory bank| 75.0±1.2
>
> We show that vanilla IGSD performs well and can be improved slightly with special negative mining mechanisms.
>
> Q2: In Section 2.1, the notation for augmentations. Why are the graphs G_L attached without labels after being augmented?
>
> A2: Thanks for pointing out the typo, we correct the notation. Graphs G_L will be attached with labels after being augmented since we perform label-preserving data augmentation.
>
> Q3: In Section 2.3, authors firstly use PPR to augment node features, then randomly remove edges to create a corrupted graph. According to the description, the question is how many views will be used in the following sections? I guess that the graph feature from the original graph will be fed to the student network, and the augmented corrupted graph will be fed to the teacher network.
>
> A3: Thanks for pointing out the confusion of description and we clarify it in the latest draft. Indeed, random and graph diffusion are two different kinds of data augmentation methods rather than a two-stage procedure. (Randomly corrupting graph features serves as another augmentation choice to validate the robustness of models on different augmentation strategies.) We employ one of these two choices to get an augmented view. In our work, the augmented view and original view are passed interchangeably to the teacher and the student network as in Equation 2.
>
> Q4: The proposed method contains an encoder, projector, and predictor. The question is why do we need a projector g to get a higher dimension z? Does it have a big influence on the performance? Could you please give the complete definition of function g and h?
>
> A4: While we could directly predict the representation y and not a projection z, previous contrastive learning work [1] in the image domain have empirically shown that using this projection improves performance. Per your question, we also further investigate the performance with and without the projector on 4 datasets:
>
> |Dataset | MUTAG| IMDB-B | IMDB-M | PTC
>
> |IGSD | 90.2 ± 0.7 | 74.7 ± 0.6 | 51.5 ± 0.3 | 61.4 ± 1.7
>
> |IGSD w/o projector | 87.68 ± 0.9 | 74.2 ± 0.6 | 51.1 ± 0.6 | 56.7 ± 0.9
>
> Results above show that dropping the projector degrades the performance, which indicates the necessity of a projector.
> The project hidden size is treated as a hyper-parameter over {1024, 2048}. To investigate its influence on model performance, we fix the size of layers in encoders so that their output size is always 512. Then we conducted ablation experiments on different size of the projection head on IMDB-B with the following results:
>
> |Size | 218 | 512 | 1024 | 2048
>
> |Acc  | 74.5±0.6 | 74.8±0.4 | 74.7±0.6 | 74.9 ± 0.8
>
> In general, the performance is insensitive to the projection size while a larger projection size could slightly improve the unsupervised learning performance.
> Thanks for pointing the definition out. We give a clear definition of function g, h and L^{G_i, G_j}, the procedure of the contrastive learning process in section 3.1 in the latest draft.
>
> Q5: The definition of L^{con} in Equation 2 is for positive samples extracted from the same graph G_i. However, the complete unsupervised loss needs negative samples G_j. Could you please also give the definition for L^{G_i, G_j}?
>
> A5: Thanks for pointing out the confusion. We change the equation to be the definition of L^{G_i, G_j}, so that L^{G_i, G_i} will be a special case that also obeys Equation 2.

---

> > ### Author Response · Authors · 2020-11-15
> > **Response to the AnonReviewer4 Cont.**
> >
> > Q6: The overall loss consists of supervised and unsupervised loss. The L^{Sup} has conflict to the first term in Equation 7. Both of them use labels but it's difficult to tell which one should be aligned with the supervised loss shown in the ablation study (Table 2). The SupCon has never been shown in the main content before. Please pay attention to make it clear.
> >
> > A6: Thanks for pointing it out. We change the L^{Sup} to be L^{Supcon} for clarity. Note that in the semi-supervised settings, we always include standard supervised loss like cross-entropy or mean square error as described in the text right above the Equation 7. Our ablation study aims to show the supervised contrastive loss can provide extra benefits with standard supervised training.
> >
> > [1] Chen et al., A Simple Framework for Contrastive Learning of Visual Representations. ICML 2020.
> >
> > [2] He et al., Momentum Contrast for Unsupervised Visual Representation Learning. CVPR 2020.

---

### Official Review · AnonReviewer1 · 2020-10-31
**Motivation of key elements and significance of the idea are unclear**

**Rating:** 5
**Confidence:** 5

**Review:**

Summary:

This paper proposes a self-distillation based graph augmentation mechanism to alleviate the drawbacks of existing MI based models w.r.t. their high dependency towards negative sampling. Quantitatively the proposed model achieves encouraging results. However it would have been better if the system designs and significant difference of IGSD from existing work are discussed.


Strength:

- This work has clearly discussed a drawback of existing unsupervised MI based models which is the leading approach in graph classification
- They propose a mechanism to address this issue with satisfiable quantitative results on unsupervised setting and extended semi-supervised setting with self-training also supported quantitatively.
- Paper is clear in general, with a clear research problem, proposes mechanism for unsupervised/semi-supervised graph representation domain and encouraging quantitative results.

Weakness:

-There is a lack of qualitative analysis and discussion of the proposed method.
-In Section 4.3 "Performance with different amount of negative pairs", it is not clear the reasoning of the provided observation from Figure 3a.
-It is not clear the motivation behind selecting a teacher-student network for obtaining different views of the graph. These networks are normally used for knowledge transfer, but here used for contrastive learning. How is this more beneficial than an ensemble model w/o knowledge transfer step of Eq. 3.
-The core difference of IGSD from CMC-graph is that CMC uses MI based contrastive between local patch representation and graph rep, while IGSD uses L2 based contrastive between 2 graph representations. Input, Encoders and projections are the same for both architectures. It could be useful to add some analysis to discuss these differences and their contributions to clearly understand the significance of IGSD.
-This paper seems to have state-of-the-art results (although it is based on graph kernel). Why the results are not included?

Convolutional Kernel Networks for Graph-Structured Data, ICML-2020



=======================

after rebuttal:


I thank the authors for the response. I still have concerns re their comparison with GCKN (ICML-2020). The reproduced results by the authors are different significantly from the published one in Table 1 of GCKN paper (~5%). E.g. MUTAG is 92.8 in original paper, but the authors report 87.2 for GCKN. The difference is significant.

Therefore, I will keep my original rating.

---

> ### Author Response · Authors · 2020-11-15
> **Response to the AnonReviewer1**
>
> Thanks a lot for your constructive comments! We have conducted more experiments and revised the paper according to your suggestions. The responses to some of your questions are listed below:
>
> Q1: Motivations & Differences of IGSD from existing work.
>
> A1: Note that we discuss the differences of IGSD with existing work in the introduction, related work section and appendix. In the related work, we categories the contrastive learning into context-instance contrast and context-context contrast [1]. Most graph representation learning works focus on context-instance contrast like MI-based models while IGSD belongs to context-context contrast.
>
> In the unsupervised learning part, the introduction of contrastive learning is well-motivated since it avoids hand-crafting domain-specific graph kernels by bringing closer samples from the same instance and separating samples from different instances. It doesn’t suffer from scalability and huge memory consumption issues of graph kernels. Specifically, we approach the graph-level representation learning problem via instance-wise contrastive learning, which is under-explored. With instance-wise discrimination, we are able to alleviate the drawbacks of related works like MI based models.
>
> For semi-supervised learning, we show that task-agnostic self-distillation would benefit semi-supervised graph-level representation learning by introducing a supervised contrastive loss. By contrast, defining such kind of loss to alleviate biased sampling problems is infeasible for existing works like InfoGraph since subgraphs don’t contain information about graph categories. Besides, we develop a self-training algorithm based on the supervised contrastive loss to leverage the information from pseudo-labels with high confidence. We discuss the benefits of it and the distinctions from conventional self-training algorithms in the last paragraph of sec 4.3.
>
> Q2: In Section 4.3 "Performance with different amounts of negative pairs", it is not clear the reasoning of the provided observation from Figure 3a.
>
> A2: Note that the scale of the vertical axis in Fig. 3a is smaller than that of Fig. 2 in the Appendix of CMC-Graph. The performance of CMC-Graph with respect to Batch Size (Negative Samples)  in IMDB-BIN dataset in our paper corresponds to that in original CMC-Graph paper, which both vary slightly as batch size increases. The observation of the increasing performance gain indicates IGSD is better at leveraging negative pairs for learning effective representations than CMC-Graph.
>
> Q3: It is not clear the motivation behind selecting a teacher-student network for obtaining different views of the graph. These networks are normally used for knowledge transfer, but here used for contrastive learning. How is this more beneficial than an ensemble model w/o knowledge transfer step of Eq. 3.
>
> A3: We note that different views of graphs are obtained via data augmentation rather than by teacher-student network.
> Our intuition to use teacher-student networks is that the EMA teacher can distill crucial knowledge to the current model. Such an iterative self-distillation process can (1) effectively distill important knowledge learned in the past to itself, and (2) serve as a regularization to stabilize the training. Our motivation is also validated by [4], in the sense that EMA outperforms ensembling in $\Pi$ and Mean Teacher models.
> The motivation of the teacher-student model is to introduce a slow-moving average teacher network that measures consistency against a student one, thus providing a consistency-based training paradigm where two networks can mutually enhance each other [4]. In this way, we don’t need extra designs for MI estimator, subgraph sampling strategies and negative sample mining approaches like increasing batch size dramatically [2] or keeping a memory bank [3].

---

> > ### Author Response · Authors · 2020-11-15
> > **Response to the AnonReviewer1 Cont.**
> >
> > Q4:The core difference of IGSD from CMC-graph is that CMC uses MI based contrastive between local patch representation and graph rep, while IGSD uses L2 based contrastive between 2 graph representations. Input, Encoders and projections are the same for both architectures. It could be useful to add some analysis to discuss these differences and their contributions to clearly understand the significance of IGSD.
> >
> > A4: Note that we include the discussions on the differences and motivations in the 2nd, 3rd, 4th paragraph in the introduction section. Although we keep only the encoders the same with baselines for fair comparisons, the components and training paradigms of IGSD and CMC-Graph are substantially different:
> >
> > CMC-Graph: Two networks in CMC-graph are symmetrical. Two different MLPs are used for encoding the local and global representations, respectively. Then an additional discriminator is used for estimating the mutual information of global and local embeddings for both positive and negative pairs following the InfoMax principle.
> >
> > IGSD: Two networks in IGSD are asymmetric. MLPs are employed as projectors to attain graph representations via ReLU nonlinearity, defining a consistency loss in Equation 2 for training. The student network is trainable with L2 norm while the teacher network is updated with EMA. The contribution of the design is that two networks can mutually enhance each other iteratively.
> >
> > Q5:This paper seems to have state-of-the-art results (although it is based on graph kernel). Why are the results not included? Convolutional Kernel Networks for Graph-Structured Data, ICML-2020
> >
> > A5: Thanks for pointing out this related work. Note that one motivation of IGSD is using contrastive graph representation learning to alleviate the flaws of graph kernels and we have already included 6 representative graph kernel baselines. The setting in original GCKN paper is different from ours and other contrastive learning works [5, 6]: GCKN uses different train-test splits for nested 10-fold cross validation with $C$ parameter of SVC is in 1/n × np.logspace(-3, 4, 60), requiring 10 times more computation.
> >
> > We conducted experiments using Stratified10fold cross validation with $C$ parameters in SVC in {1e-3, 1e-2, . . . , 1e2, 1e3} to compare with GCKN with results as follows:
> >
> > Dataset | MUTAG | IMDB-B | IMDB-M | NCI1 | COLLAB | PTC
> >
> > Ours | 90.2±0.7 | 74.7±0.6 | 51.5±0.3 | 75.4±0.3 | 70.4±1.1 | 61.4±1.7
> >
> > GCKN | 87.2±6.8 | 70.5±3.1 | 50.8±0.8 | 70.6±2.0 | 54.3±1.0 | 58.4±7.6
> >
> > In the same setting, GCKN outperforms other graph kernels but IGSD performs consistently better than GCKN in all datasets, especially in large-scale ones like NCI1 and COLLAB.
> >
> > Despite the effectiveness of GCKN in the unsupervised tasks, there is no free lunch:
> > For COLLAB, GCKN learns the anchor points $Z_j$ for each layer by K-means over 300000 extracted paths from each training fold, which requires huge memory and computation consumption. As indicated in the original paper of GCKN, GCKN’s major limitation is the exponential complexity of (long) path enumeration, which requires to compute the feature map $\phi$ and prevents us from using long-length extraction as soon as the graph is dense (which corresponds to the inferior performance in COLLAB and NCI1 datasets).
> > Besides, as a multilayer kernel for graphs based on paths, GCKN also needs hand-crafted (differentiable) kernel functions (such as the exponential function described in the “Interpretation as a GNN” subsection of section 3 in the original paper), which is one of the motivations for adopting end-to-end or iterative contrastive learning (with common GNNs in the loop).
> > The process of GCKN for attaining node representations is similar to that of message-passing process in GNNs: path extraction, kernel mapping and path aggregation, which only serves as the role of encoder in our model.
> >
> >
> > [1] Liu et al., Self-supervised Learning: Generative or Contrastive. Arxiv.
> >
> > [2] Chen et al., A Simple Framework for Contrastive Learning of Visual Representations. ICML 2020.
> >
> > [3] He et al., Momentum Contrast for Unsupervised Visual Representation Learning. CVPR 2020.
> >
> > [4] Athiwaratkun et al., There Are Many Consistent Explanations of Unlabeled Data: Why You Should Average. ICLR 2019.
> >
> > [5] Sun et al., Infograph: Unsupervised and semi-supervised graph-level representation learning via mutual information maximization. ICLR 2019.
> >
> > [6] Hassani et al., Contrastive Multi-View Representation Learning on Graphs. ICML 2020.

---

### Author Response · Authors · 2020-11-16
**Submission Revision 1: Summary of Changes:**

We would like to thank all the reviewers for your constructive reviews. We’ve revised the paper accordingly. The revised part is highlighted as blue. Specifically, we have made the following changes:

1. We conduct additional experiments to compare our proposed method IGSD with two baselines GCC and GCKN. Results are shown in Table 1, showing that IGSD can still get competitive results on most datasets we use.
2. We conduct ablation experiments to show the effect of projections on the performance of IGSD with results in Appendix A.3.
3. We move the related work section after the introduction, which helps readers better understand the motivations and differences with existing works on graph representation learning.
4. In sec 4.1, We discuss the difference between the EMA update in our work and that in existing work MoCo in more detail.
5. For consistency, we change all $\tilde{G}$ to be $G^\prime$ and $\mathcal{L}^{sup}$ to be $\mathcal{L}^{supcon}$.
6. We revise the second paragraph in the sec 3.1 to include the descriptions of contrastive learning procedures with both positive and negative pairs. And we change the notation of $v$ to be $G$ in Figure 1 accordingly. We give specific definitions of f, h and g in sec 4.1.
7. We revise the Eq. 2 from $\mathcal{L}^{con}(G_i, G_i)$ to be a more general form $\mathcal{L}^{con}(G_i, G_j)$, which is applicable for both positive and negative pairs.
8. We add some statistics of the QM9 dataset: the amount of atoms in each molecule is 9 in the **datasets** part in section 4.1. And additional details of the QM9 dataset can be found in the Appendix C of the InfoGraph paper.

---

### Decision · Program_Chairs · 2021-01-07
**Final Decision**

**Decision:**

Reject

**Comment:**

This paper proposes an unsupervised graph learning method [Iterative Graph Self-Distillation (IGSD)] by iteratively performing self-distillation to contrast graph pairs under different augmented views. This idea is then extended to semi-supervised setting where via a supervised contrastive loss and self-training. The method is empirically evaluated on some semi-supervised graph classification and molecular property prediction tasks, and has achieved promising results.

Reviewers agree that the method is interesting and the paper is well-written. The biggest concern from reviewers related to experimental evaluations of the method. The authors responded to this and included additional experiments. Although the reviewers appreciate the provided results and explanations, at the end they were not convinced about the empirical assessments. In particular, R1's post rebuttal comment indicates concerns about the reported performance of GCKN, which is different from the published one in Table 1 of GCKN paper. I encourage authors to improve on these experimental discrepancies and resubmit.